# Assessment of Protein Quality and Nutritional Characteristics of Commonly Consumed Pulses in the Caribbean Diet by Different In Vitro Assays

**DOI:** 10.3390/foods14020283

**Published:** 2025-01-16

**Authors:** Daniel J. Thomas, Zhanhui Lu, Yolanda Brummer, Yan Zhu, Ronghua Liu, Lili Mats, Rong Tsao, D. Dan Ramdath, Matthew G. Nosworthy

**Affiliations:** 1Caribbean Institute for Health Research, The University of the West Indies, Mona, Kingston 7, Jamaica; 2Guelph Research and Development Centre, Agriculture and Agri-Food Canada, Guelph, ON N1G 5C9, Canada; zhanhui.lu@agr.gc.ca (Z.L.); yolanda.brummer@agr.gc.ca (Y.B.); yan.zhu@agr.gc.ca (Y.Z.); ronghua.liu@agr.gc.ca (R.L.); lili.mats@agr.gc.ca (L.M.); rong.cao@agr.gc.ca (R.T.); dan.ramdath@agr.gc.ca (D.D.R.); 3College of Pharmacy and Nutrition, University of Saskatchewan, Saskatoon, SK S7N 5E5, Canada

**Keywords:** legumes, protein quality, in vitro protein digestibility, PDCAAS, PER, phenolics, starch

## Abstract

Pulses, the dried seeds of leguminous plants, form an important part of the diets of many cultures, including Caribbean cuisine, and are a rich source of protein, carbohydrates, and antioxidants while being low in fats. This study examined the effect of a traditional home-cooking method on the nutritional characteristics of pulses commonly consumed in the Caribbean: red kidney beans and cranberry beans (*Phaseolus vulgaris* L.), cowpeas (*Vigna unguiculata* L.), and pigeon peas (*Cajanus cajan* L.). Protein quality, determined via three in vitro protein digestibility methods, starch, and phenolic content were determined in pre- and post-cooked samples using established methods. Pulses contained 20–26% protein, and cooking improved protein digestibility on average by 14.0 ± 2.5% (*p* < 0.05). However, notable differences in digestibility were observed: it was higher in static assays (pH-Drop and pH-Stat) than in the two-step digestibility assay. Average protein digestibility-corrected amino acid score (IVPDCAAS) among cooked pulses was 0.81 ± 0.14, with the highest in cranberry bean (0.82) and cowpea (0.88). Cooking modified pulse starch profiles by increasing total digestible starch. However, resistant starch and slowly digestible starch fractions accounted for approximately 20–25% of total cooked starch content. While total phenolic content (TPC) and antioxidant activity were reduced with cooking, they were within expected ranges for cooked pulse flours; however, they were higher in bean (*P. vulgaris*) varieties than cowpea and pigeon pea. These findings support the promotion of increased pulse consumption in Caribbean diets. Home cooking is a simple method to enhance pulse protein quality through enhancing digestibility; however, in vitro protein digestibility assays may require further standardization.

## 1. Introduction

Pulses, the dried seeds of legumes, serve as a key ingredient in the cuisine of many cultures, including that of the Caribbean, which has historically been marked by Afro-Caribbean and Indo-Caribbean influences. Commonly consumed pulses of Afro-Caribbean origin include many varieties of the common bean (*Phaseolus vulgaris* L.), such as red kidney bean, black bean, and pinto bean, as well as peas such as pigeon pea (*Cajanus cajan* L.) and cowpea (*Vigna unguiculata* L.) [1]. Similarly, there is an Indo-Caribbean influence as evident by the inclusion of chickpeas (*Cicer arietinum* L.), split peas (*Pisum sativum* L.), and lentils (*Lens culinaris* L.) in Caribbean diets [2]. In Jamaica pulses are often consumed multiple times weekly [3], traditionally prepared by boiling or pressure cooking until tender for consumption in stews or soups and often paired with grains such as rice [1,4]. There is, however, limited data on the nutritional quality of commonly consumed pulses in the Caribbean context, particularly the pulse varieties of Afro-Caribbean origin.

While pulses are rich dietary sources of protein, their amino acid profile is not optimal for human protein requirements, since they are typically limiting in the sulfur amino acids (SAA) methionine and cysteine [5]. Pulses also contain antinutritional factors, such as tannins, phytic acid, and protease inhibitors [6], which may influence their digestibility by inhibition of protease enzymatic activity. Although there are multiple metrics for assessing protein nutritional quality, the Protein Digestibility Corrected Amino Acid Score (PDCAAS) is used for protein quality assessment by many regulatory agencies [7]. This method relies on the product of an amino acid score, generated by comparing the amino acid composition to a reference pattern of human requirements, and the net protein digestibility [8]. Currently, regulatory guidelines in North America require in vivo assessment of protein digestibility via animal experimentation; however, there is considerable advocacy from consumers and product developers to adopt in vitro protein digestibility (IVPD) methods to reduce reliance upon animal models. There are multiple different methods in use to determine IVPD, including static mono-compartmental models (pH-Drop and pH-Stat) and more complex assays such as computer-controlled, dynamic, multicompartmental models (TIMs) [9]. Results generated from these IVPD can then be used to generate an in vitro PDCAAS (IVPDCAAS), which has been demonstrated to be well correlated with data generated in vivo, with R^2^ values between 0.93 and 0.99 for various protein sources [10,11,12,13]. For this reason, it is important to utilize these methods to evaluate the protein quality of beans to further validate their use in nutritional quality assessment.

Pulses also serve as nutritious sources of carbohydrates in the form of starch and fiber, in addition to being ready sources of dietary protein [14]. Of particular importance, pulse carbohydrates are high in slowly digestible starch (SDS), resistant starch (RS), and fiber when compared with other starchy food sources [15,16], and these have been shown to promote cardiometabolic health and integrity of the gut microbiome [17,18]. Additionally, the antioxidant capacity of pulses, based on their phenolic content and other bioactive compounds, is of particular interest to both consumers and product developers due to their health benefits [19,20]. Phenolics influence a wide range of biological functions, including anti-inflammatory, anti-cancer, hypolipidemic, hypoglycemic, and enzyme inhibitory activities [21]. It is therefore important to understand the nutritional characteristics and health attributes of pulses prepared by common domestic preparation methods.

This study was undertaken to assess protein quality and other nutritional properties of milled flours of pulses commonly consumed in the Caribbean diet as prepared by a customary home-cooking method. Of significance, protein quality scores as determined by IVPDCAAS were assessed using three established static in vitro protein digestibility assays, two single-step models (pH-Drop and pH-Stat) and a two-step assay. Multiple regression equations have been published to derive IVPDCAAS via the pH-Drop and pH-Stat assays, and protein quality scores obtained were compared to determine differences based on assay and regression model. Similarly, starch profile was analyzed by two established starch digestibility assays, again to observe differences between analytical methods. Phenolic content and antioxidant capacity of pulse flours were evaluated to better understand the role of pulse diets in reducing the incidence of chronic diseases. As such, this work offers a comprehensive examination of the effect of cooking on both the nutritional and health attributes of commonly consumed pulses in the Caribbean diet.

## 2. Materials and Methods

### 2.1. Chemicals and Reagents

Standard reference materials, including L-ascorbic acid, gallic acid, fluorescein, 2,2-diphenyl-1-picrylhydrazyl (DPPH), Trolox, and 2,20-azobis-(2-amidinopropane) dihydrochloride (AAPH), were purchased from Sigma-Aldrich Co. (St. Louis, MO, USA). Sodium acetate, monobasic sodium phosphate, dibasic sodium phosphate, and HPLC-grade solvents, including methanol (MeOH) and formic acid, were purchased from Caledon Laboratories (Georgetown, ON, Canada) or Sigma-Aldrich (Oakville, ON, Canada). Ferric chloride hexahydrate, sodium acetate, 1,3,5-tri(2-pyridyl)-2,4,6-triazine (TPTZ), 2,2-diphenyl-1-picrylhydrazyl (DPPH), Folin–Ciocalteu’s phenol reagent, NIST 3234, pepsin (porcine gastric mucosa, P7000, ≥250 units/mg), pancreatin (porcine pancreas, P1750; 4 × USP specifications), potassium phosphate buffer (0.2 M, pH 7.0), chloramphenicol, trichloroacetic acid, o-phthaldialdehyde (OPA), sodium borate (Borax), β-mercaptoethanol, L-leucine, diethyl ether–ethyl acetate, hydrochloric acid (HCl), and sodium hydroxide (NaOH) were purchased from Sigma-Aldrich (Oakville, ON, Canada). Acids were of ACS (American Chemical Society, Washington, DC, USA) grade, and those not listed above were supplied by Fisher Scientific (Nepean, ON, Canada). Thermostable α-amylase, amyloglucosidase, and GOPOD reagent were from Megazyme (Bray, Ireland).

### 2.2. Sample Procurement and Preparation of Cooked Flours

Samples of red kidney beans (*Phaseolus vulgaris* L.), Miss Kelly Kidney Beans, a Jamaican variety of red kidney beans, and cranberry beans (*P. vulgaris* L.) were obtained from HiLo supermarkets, Kingston, Jamaica. Cowpeas, or black-eyed peas (*Vigna unguiculata* L.), and pigeon peas (*Cajanus cajan* L.) were procured from No Frills Supermarket, Guelph, Ontario, Canada.

Pulses were prepared according to a standard home-cooking method [22]. Samples were soaked overnight at room temperature in distilled water at a 4:1 ratio of water to sample. They were then rinsed 3 times with distilled water, transferred to 1 L stainless steel pots, and distilled water was added at a 4:1 ratio of water to pulse seed dry weight. Pulses were brought to a boil and maintained until cooked, approximately 25–35 min (indicated by 4 out of 5 pulse seeds being easily compressible between thumb and index finger). Samples were subsequently drained, cooled, and air-dried prior to freeze-drying (Labconco, Freezone 12L) at 0.120 mba, −40 °C for approximately 72 h. Cooked and unprocessed samples were milled in an UDY cyclone sample mill (Model 3010_030, Fort Collins, CO, USA) through a 1 mm sieve. Milled flours were stored at −20 °C until further analysis.

### 2.3. Proximate Analysis

Protein and fat contents were analyzed at the Central Analytical laboratory (University of Arkansas, 1260 W. Maple, Fayetteville, AR 72701, USA), using AOAC 968.06-1969 [23] and AOCS Standard Procedure Am 5-04 [24], respectively. Moisture content of milled pulse flours was determined using an Isotemp Vacuum Oven Model 280A (Fisher Scientific Company, Nepean, ON, Canada) based on the AACC method 44-17 [22].

### 2.4. Starch Analyses

Starch content of the pulses was determined as previously described [25]. In brief, total starch of unprocessed and cooked pulses was measured using the dimethyl sulfoxide (DMSO) method of the Megazyme (Megazyme International Ireland Ltd., Bray, Co., Wicklow, Ireland) kit K-TSTA, according to AOAC method 996.11 [26]. Starch digestibility was assessed using the Megazyme kit K-DSTRS, and resistant starch was also determined via the Megazyme kit K-RSTAR according to AOAC Method 2002.02 [27].

### 2.5. Amino Acid Analysis

A complete amino acid profile was determined using the AOAC Official Method 982.30, with isolation of sulfur amino acids by performic acid oxidation (AOAC Official Method 994.12) and tryptophan being determined by alkaline hydrolysis (AOAC Official Method 988.15) [26]. Available lysine was determined by AOAC Official Method 975.44 [26]. These analyses were completed at the Agriculture Laboratory Analytical Services of the University of Missouri, 700 Hitt Street, Columbia, MO 65211.

### 2.6. In Vitro Protein Digestibility by pH-Drop and pH-Stat Digestibility Assays

The pH-Drop (PHD) assay and regression equations as outlined by Pederson et al. [10], Hsu et al. [11], and Tinus et al. [12], and the pH-Stat (PHS) assay and related regressions equation, as described by Pederson et al. [13], were used to determine IVPD of unprocessed and cooked pulse flours. A casein control was included in each experimental run as a control sample.

Regression equations used to derive IVPD by the PHD assay were as follows:(1)$$PHD1=74.51+\left(22.51\times {∆pH}_{10\;min}\right)\;$$(2)$$PHD2=73.75+\left(22.56\times {∆pH}_{10\;min}\right)\;$$(3)$$PHD3=76.97+\left(19.16\times {∆pH}_{10\;min}\right)\;$$(4)$$PHD4=65.66+\left(18.10\times {∆pH}_{10\;min}\right)$$
where ΔpH is the change in pH in 10 min from the initial pH of approximately 8.0.

Regression equations used to derive IVPD by the PHS assay were as follows:(5)PHS1=76.14+(47.77×∑mL 0.1N NaOH)(6)PHS2=78.61+(40.29×∑mL 0.1N NaOH)
where Σ mL is the total volume of 0.1 N NaOH titrated over 10 min to maintain the solution at a pH of 8.0.

### 2.7. In Vitro Protein Digestibility by Two-Step Static Enzymatic Digestibility Assay

The two-step digestibility (TSD) assay was conducted as previously described by Franczyk [28]. A casein control was included in each experimental run as a control sample.

### 2.8. In Vitro Protein Digestibility Corrected Amino Acid Score

The in vitro protein digestibility corrected amino acid score (IVPDCAAS) was calculated as a product of the amino acid score (AAS) and IVPD% as determined by each digestibility assay and applicable regression equations (PHD, regression Equations (1)–(4); PHS, regression Equations (5) and (6); and TSD).IVPDCAAS (%) = IVPD% × Amino Acid Score (AAS) 
where the AAS is determined by comparing the amino acid composition of the protein source to the reference pattern recommended by the FAO/WHO, the amino acid requirements of a 2–5-year-old child [8], with the first limiting amino acid conferring the AAS.

### 2.9. Determination of Total Polyphenol Content (TPC) and Antioxidant Activity of Pulse Flours

Extraction of free and conjugated phenolics from pulse flours was carried out as previously described by Chen et al. [29]. Bound phenolics were extracted by methods as previously described by Sun et al. [30]. TPC, DPPH, FRAP, and ORAC assays were determined using methods as previously described by Li et al. and Zhang et al. [31,32].

### 2.10. Statistics

Data analysis was performed using SPSS Statistics Software Version 22. Data are presented as mean ± SD. Differences in proximate composition and in vitro protein and starch digestibility were compared by one-way analysis of variance (ANOVA) with Tukey’s HSD multiple-comparison test. Differences among IVPD assays were determined by the Robust test for equality of means with post hoc analysis by the Games–Howell test. Differences between resistant starch values obtained by K-RSTAR vs. K-DSTRS were determined by paired *t*-test. A *p*-value < 0.05 was chosen to indicate statistical significance.

## 3. Results and Discussion

### 3.1. Proximate Composition

The proximate composition expressed as percentage dry weight mass of the unprocessed and cooked samples is presented in Table 1. Cooked pulse samples were freeze-dried in preparation for milling, which resulted in a higher dry matter percentage of cooked pulse flours when compared to unprocessed flours that were not freeze-dried.

There was no difference in the protein content of unprocessed and cooked pulse flours on a dry matter basis. Protein content of cooked pulse flours ranged from 20.5 to 26.9%, lowest in pigeon pea and highest in cowpea (Table 1). This is similar to previously published data where the protein content of beans ranged between 21.4 and 23.6% [33], cowpeas 16–31% [34], and pigeon peas 16–24% [34]. Fat and starch proximate content were higher in cooked samples, which may be due to loss of other components of pulse during soaking and the cooking process [35,36]. Fat content of cooked pulses ranged from 1.73 to 2.41% (Table 1). Of the two pulse categories, cooked peas had overall higher fat content (cowpea 2.03%, pigeon pea 2.41%), while among bean varieties, both kidney bean varieties studied had significantly lower fat content (1.73%, 1.84%) than cranberry beans (2.25%) (*p* < 0.05). This is similar to published values for pulse fat content [34,37,38,39,40], although even greater fat content has been demonstrated in studies on pigeon pea varieties in Nigeria (4.5%) [41] and India (3.92%) [42].

The starch content of cooked pulses ranged from 41.6 to 51.1% (Table 1). Similar to fat, pea varieties had the highest starch percentage content, of which cowpea had the highest (51.1%) among pulse varieties. Again, cooked cranberry beans had significantly higher starch content (45.1%) than the two kidney bean varieties (41.6% and 42.6%) (*p* < 0.05). While there was a wide variation in starch content between pulse classes, the range is within that of published values (41–51%) for cooked pulses [15,37,43].

Proximate composition may vary widely between varieties of pulse grains, influenced by both genetic and environmental factors [44]. Within pulse species *P. vulgaris* L., there were notable differences in protein, fat, and starch contents, with unprocessed red kidney beans demonstrating significantly higher protein content and lower fat content than cranberry and Miss Kelly, a Jamaican-grown variety of kidney bean (Table 1). Similarly, within the *Faboideae* subfamily of peas, there was a significant difference in macronutrient values between cowpea and pigeon pea varieties. This speaks to the effect of genetic variation and environmental influence, such as geographic location, soil type, and weather conditions, on the nutritional characteristics of pulses [39,40].

A negative correlation between starch and oil contents and protein content has been reported in the literature [34], and the trends in the present study are in agreement. Overall, proximate values for protein, fat, and starch content for pulses studied are consistent with ranges in the literature.

### 3.2. Protein Digestibility

Results from in vitro digestibility are presented in Table 2. Two multi-enzyme cocktail approaches, PHD and PHS, were compared to a static two-step digest (TSD). Protein digestibility trends were similar between multi-enzyme cocktail approaches, with IVPD values for unprocessed pulses ranging from 73 to 87% and 77–83% by PHD and PHS assays, respectively. Similarly, for cooked pulses IVPD ranged from 83 to 99% and 91–98% for PHD and PHS, respectively. When examined separately, IVPD for cooked pulses derived by regression equations PHD1-3 (95–99%) and PHS1-2 (91–98%) were significantly higher than PHD4 (82–85%) (*p* < 0.05). IVPD from PHD and PHS assays was significantly higher (*p* < 0.05) than the static two-step digest, whose IVPD ranged from 45 to 59% for unprocessed and 61–73% for cooked pulses.

While these simple static PHS and PHD methods provide a convenient method of determining IVPD from ΔpH or ∑ mL 0.1N NaOH, there are inherent limitations based on the regression equations used to derive the digestibility coefficients. The multi-enzyme cocktail approaches rely on regression equations that correlate a ΔpH or ∑ mL 0.1 M NaOH to a level of protein digestibility based on true protein digestibility (TPD) derived in rats (FAO/WHO, 1991). These equations can algebraically be represented as y = mx + k, where k is a constant and x is the ΔpH or ∑ mL 0.1 M NaOH. Of the 3 Pedersen IVPD equations (PHD1, 2, and 3), PHD1 has been used as a general IVPD equation, PHD2 is indicated for all protein sources except eggs and non-fat dry milk, and PHD3 has been indicated for plant proteins and caseins [10]. PHS1 has been used for general protein digestibility, and PHS2 has been specified for plant-sourced proteins [13].

Examining these regression formulae, due to the given constant (k), IVPD will have a minimum value of about 73–76% (PHD1-3) or 66% (PHD4) and 76–78% (PHS1-2) with no protein digestion, and consequently, no change in pH, or addition of NaOH to the digesta. Hence, for the same ΔpH, digestibility values derived from equations PHD 1-3 will be on average 8–10% higher than PHD4, as was seen in the present study when digestibility values derived by the four PHD equations were compared. This disparity in digestibility is significant, and further work is needed to confirm the most appropriate regression equations using the PHD method with different protein sources. It should also be noted that while these equations are bound by a lower limit of digestibility, mathematically a digestibility coefficient greater than 100% may be derived by these regression equations depending on ΔpH_10 min_ or ∑ mL 0.1N NaOH [12].

In vivo and in vitro protein digestibility studies have varied in their derived digestibility ranges. The IVPD PHS assay was used in a collaborative study between six laboratories in four countries (USA, Canada, Denmark, and the Netherlands) [45] in which the digestibility of 17 protein sources was analyzed. Among these were soy isolate, pea concentrate, chickpeas (canned), and pinto beans (canned).

This collaborative study reported IVPD values of 99.0, 97.6, 93.8, and 93.7%, respectively. TPD of cooked kidney beans measured in rat and pig models and one human isotope study has ranged from 76 to 82% [46,47,48,49,50,51], while IVPD was approximately 82% [33]. However, a study by Khattab et al. using PHD4 obtained IVPD values of 87–94% for red kidney beans and 97–98% for cooked cowpea varieties [52]. IVPD of cooked cowpeas determined previously by static gastric–intestinal assay was 29.4–44.1% [53]. However, IVPD was found to be 68–72% in Ethiopian-grown cowpeas [54] and 83–86% in an Australian study using equation PHD4 [12]. Singh et al. reported cooked pigeon pea protein digestibility of 71–88% by an in vivo rat bioassay [55] but IVPD of 41–68% [56]. Other reported average digestibility values have been 79% and 83% (cowpea), 59% and 60% (pigeon pea), and 56% and 80% (common bean) for unprocessed and cooked pulses, respectively [57].

Possible explanations for the varying digestibility coefficients may be attributed to interspecies variations, genetic variations that may affect protein fraction percentages [58], the level of anti-nutritional factors such as anti-trypsin activity [57], and also the effect of mill type and particle size [12,39]. Of the multi-enzyme assays conducted in the present study, IVPD values derived by regression equation PHD4 were similar to the in vivo TPD and IVPD ranges (75–82%) reported by others [33,46,49,50,59].

The static TSD assay has no pre-set or implicit lower limit, as is the case with the regression equations employed to determine IVPD from PHD and PHS assays. Hence, a wider range of protein digestibility values can be determined, which may make it a more useful assay, particularly when used to analyze proteins of low physiological digestibility. As evidenced in the present study, average digestibility by TSD ranged from a low of 42% in unprocessed kidney beans to a high of 72% in cooked cranberry beans (Table 2). Maximum IVPD for cooked pulses by this static method was within the lower range for expected digestibility of pulse proteins. However, there was high variability in IVPD between TSD runs, particularly in red kidney bean samples, which had coefficients of variation of approximately 25% and 33% in respective unprocessed and cooked samples. This indicates the need for further refinement of this method to ensure homogeneity in the sampling of pulse flour ingredients as well as consistency within methodological steps. While this assay is promising, continued modification and validation of this method are required to increase reliability.

### 3.3. In Vitro Protein Digestibility Corrected Amino Acid Score

Amino acid composition is presented in Table 3, with the highlighted amino acid scores in Table 4, and the in vitro protein digestibility corrected amino acid score (IVPDCAAS) in Table 5. Among cooked pulses, the SAA, cysteine, and methionine were found to be limiting, with amino acid scores ranging from 0.86 to 1.02. The exception to this was the pigeon pea pulse variety for which the first limiting AA was tryptophan, with AAS of 0.71 (Table 4). While these AAS are high, they are still comparable to values in the literature. While SAA and tryptophan tend to be the lowest-scoring AAs for pulses, wide variability in AAS can be noted among studies of similar pulse varieties. Published AAS values for processed pinto beans ranged between 0.80 and 0.91 and 0.83–0.91 for processed black beans [33]. Another study that analyzed 480 samples of field peas demonstrated AAS ranging from 0.65 and 1.15, with the highest score frequency ranging between 0.85 and 1.00 [60]. Variability in digestibility coefficients derived from IVPD assays resulted in differences between IVPDCAAS by PHD, PHS, and TSD. Average IVPDCAAS derived from the five Pedersen equations for PHD and PHS were 0.68 (pigeon pea), 0.83 (red kidney bean), 0.90 (Miss Kelly), 0.94 (cranberry bean), and 1.00 (cowpeas) (Table 5). IVPDCAAS derived from PHD4 ranged from 0.59 (pigeon pea) to 0.88 (cowpea) and 0.50 (pigeon pea) to 0.70 (cranberry bean) by the TSD (Table 5). Values from these latter two approaches (PHD4 and TSD) are more similar to published PDCAAS values of 0.50–0.70 for bean cultivars [33] and 0.62–0.68 for pea cultivars [39]. This again highlights the importance of the methodology used to arrive at IVPDCAAS from the respective digestibility regression equations.

Due to the reactivity of lysine with reducing sugars (Maillard reactions) that may occur with high-heat processing methods, total lysine content may not reflect bioavailable lysine. To account for this, total and reactive lysine were measured in pulse flour samples before and after cooking. Among unprocessed flours, total lysine (g/100 g pulse flour) ranged from 1.43 g in pigeon peas to 1.89 g in red kidney beans, and in cooked flours from 1.59 g in pigeon peas to 2.05 g in cowpeas. These values are consistent with reported total lysine ranges (1.61–2.21 g) in cooked and unprocessed pulses [33,47]. Cooking did not significantly decrease the content of available lysine in pulse flour samples, with the difference between total and available lysine content ranging from 0.02 to 0.04 g lysine per 100 g pulse flour in both unprocessed and cooked flours, equating to 97.4–98.3% availability. These results are not surprising, as traditional home-cooking methods, such as boiling, apply low moist heat over short cooking times, which is less likely to induce the Maillard reaction that favors higher cooking temperatures (>120 °C) with an increased rate of reaction at increasing temperatures [61].

### 3.4. Starch Content and Profile

Additional results of the starch digestibility profile are shown in Table 6 as determined by the Megazyme kit 4-h K-DSTRS assay to directly determine RDS and SDS and RS, calculated as TS − (RDS + SDS) (labeled as RS_4h_). RS (labeled as RS_16h_) and non-RS (labeled as TDS) were also determined via the 16-h Megazyme kit K-RSTAR assay according to AOAC Method 2002.02 [27]. Starch fractions were calculated as a percentage of total starch (%TS). Trends were similar among pulse flours except cowpea, which had distinct differences in starch digestion fractions in the unprocessed state. As the two digestibility assays differ both in the amount of PPA/AMG applied and time incubated, it is not insightful to compare the RS_4h_ and RS_16h_ values of the same sample. We will focus on each protocol and discuss the differences among pulse varieties and between the treatments.

Cooking increased the overall non-RS fraction of pulse flours. RDS fractions significantly increased (*p* < 0.05) among all pulses from 3.32 to 12.86%TS in unprocessed pulses to 71.72–81.41%TS in cooked pulses. SDS in unprocessed beans ranged from 7.05 to 8.7%TS. With cooking, there was an increase in SDS fractions in beans, though not significant in red kidney beans, with an increase from 8.70 to 9.83%TS; there was a significant increase (*p* < 0.05) in cranberry beans (7.05 to 11.03%TS) and in Miss Kelly kidney beans (7.29 to 20.87%TS). There was no significant difference in SDS between unprocessed (12.98%TS) and cooked (13.42%TS) pigeon pea. Cowpea SDS fraction was significantly higher (*p* < 0.05) in unprocessed flour (40.47%TS) compared with cooked flour (12.97%TS).

RS_4h_ fractions, assessed after 4 h of starch hydrolysis (K-DSTRS), were significantly higher in all unprocessed pulse flours as compared to cooked. RS_4h_ fractions in unprocessed flours ranged from 83.7 to 89.44%TS in most pulse varieties, except cowpea flour, which was significantly lower (46.67%TS). Cooked RS_4h_ fractions ranged from 5.17 to 13.08%TS. After 16 h of hydrolysis (K-RSTAR), RS_16h_ fractions of unprocessed bean flours ranged from 87.55 to 91.83%TS, 65.9%TS for pigeon pea flour, and 2.76%TS for cowpea flour. RS_16h_ fraction was again significantly lower in cooked pulse flours, ranging between 8.73 and 9.79%TS, except cooked cowpea flour, which had a significantly lower RS_16h_ fraction (3.98%TS).

These general trends are in keeping with published data on pulse starch digestibility profiles [35,62]; however, much variation has been observed among studies [43]. Average TS and RS content of two varieties of cooked common bean flour were 36 g/100 g and 4.4 g/100 g, respectively [62], which is equivalent to 12.2% RS (%TS) and comparable to our data. Similarly, measured respective RS content in cooked cranberry bean and dark red kidney bean flour was 4.04 g/100 g and 3.59 g/100 g, while in raw beans it was 30.1 g/100 g and 29.7 g/100 g [35], which are also comparable to our results when their data are converted to a total starch basis. Chen et al. [63] reported TS of unprocessed red kidney beans to be 43.9 g/100 g, with RS being only 32.3%TS. Starch digestibility profile (%TS) from a single study on unprocessed pigeon pea flour was similar to the present study with RS 78.9, SDS 16.2, and RDS 4.2 [64]. As previously stated, these variations may be due to methodological differences, including mill type and pulse flour particle size, biological differences such as genetic or varietal/cultivar differences, as well as differing environmental conditions under which pulses were grown [12,65].

Unprocessed cowpea flour is unique in its starch digestion profile compared to other pulse flours studied, having extremely low quantities of RS (both RS_4h_ and RS_16h_) (Table 6). This finding is supported by other studies on cowpea flour. RS fractions in five varieties of unprocessed milled cowpea flour were reported by Eashwarage et al. [43] as 3.2, 3.6, 9.0, 9.1, and 9.6 g/100 g, equivalent to 7.8, 9.1, 20.9, 18.9, and 19.9%TS. Sreerama et al. [66] showed that RS content of unprocessed cowpea flour was approximately 5.2%TS. Tinus et al. [12] also demonstrated that the digestible portion of available starch of unprocessed cowpea flour tended towards 100%TS, indicating an extremely low proportion of RS. The highest reported RS content (%TS) for unprocessed cowpea flour was 28.7%TS and 14.9%TS for boiled cowpea [63]; however, sample preparation in this study was via mechanical agitation with glass beads and blending to simulate chewing, which might not destroy the food matrix as much as by milling.

K-RSTAR and K-DSTRS provide two different methods of determining resistant starch fraction, i.e., RS_16h_ and RS_4h_, respectively. K-DSTRS identifies RS over a 4-h hydrolysis assay, while K-RSTAR determines RS after 16 h of hydrolysis, which will result in a greater total digestible starch (TDS) fraction of pulse flours. RS fractions determined in unprocessed bean (*P. vulgaris*) flours studied are similar between both assays and were also significantly higher than the unprocessed cowpea and pigeon pea flours (*p* < 0.05). Unprocessed pigeon pea and cowpea flours studied had lower RS by both assays but significantly lower RS when determined after 16 h of hydrolysis (*p* < 0.05), with cowpea having extremely low RS.

Differences in starch digestibility profiles between beans and other pulse flours may be attributed to various reasons, such as supramolecular structure (packing of crystallites inside starch granules), ratio of amylose to amylopectin, fine structure of amylose, as well as surface characteristics of starch granules [67]. Published research on cowpea has demonstrated that it has a lower amylose content than other pulse starches. Additionally, cowpea appears to have a higher ratio of long-chain amylopectin when compared to other pulses [68]. Cowpea starch granules had a higher swelling volume than other pulse starches, which is linked to having lower amylose content [69]. Increased swelling of starch granules has been shown to further promote exudation or leaching of amylose [67,70]. These combinations of factors strongly indicate that, compared to other pulses, cowpea starch may have a relatively low content of amylose–lipid complexes, a form of resistant starch (RS type 5) that form with greater frequency in high amylose starches [71]. Cowpea starch may also have a different crystalline pattern from other pulse starches. X-ray diffraction of pulse starches typically demonstrates a C-type structure, which is a combination of A-type starches that are slowly digested and B-type starches that resist enzymatic digestion [67]. X-ray diffractograms demonstrate that cowpea had a slightly different crystalline structure, with 2Θ peaks not observed in other pulse starches [68], while another study by El Faki et al. [72] reported cowpea starch as A-type. Again, cowpea starch has been described as having a smaller granule size and a fine fiber matrix, making the starch light and loose [68], which may result in a reduced matrix effect, increasing starch digestibility. It is possible that pigeon pea may exhibit similar physicochemical properties to cowpea starch; however, the literature is not confirmatory.

Another possible explanation for the observed differences in starch digestibility profile may be differing ratios of various galacto-oligosaccharides (GOS), as Ramdath et al. [25] noted that this affected the composition of RS and non-RS (SDS and RDS) in lentils. It is likely that these GOS ratios are significantly different in cowpea and pigeon pea starch compared to other bean flours. This may also explain the observation of the significantly higher SDS fraction in cowpea and pigeon pea flour as compared with the other pulse varieties; however, no reports on RDS and SDS fraction of cowpea were found in the literature for comparison. Upon cooking, the RS and SDS fractions of cowpea and pigeon pea no longer differed significantly from the other pulses studied. GOS ratios may also explain the differences seen among the varieties of *P. vulgaris* L. studied, as it was noted that the proportion of non-RS content of cooked Miss Kelly kidney beans, a cultivar native to the Caribbean, had significantly higher SDS and lower RDS than Cranberry and Red Kidney bean varieties; this may reflect genetic and environmental differences within species.

On average, RS and SDS fractions in cooked pulses accounted for approximately 20–25% of total starch content. This high ratio of slowly digestible and resistant starch fractions, in addition to the high fiber content of pulses, contributes to their low glycemic index when compared to other starch-rich foods. Hence, a diet rich in pulses can be promoted as an adjunct to improving glycemic control [18,73] and offering distinct benefits in the stabilization of blood sugar levels by reducing spikes following meals [74,75,76].

### 3.5. Total Phenolic Content and Antioxidant Activity

The TPC of the extracts of different samples and their antioxidant activities measured by the FRAP, DPPH, and ORAC assays are summarized in Table 7. As expected, there was a significant loss of phenolic content and concomitant antioxidant activity post-processing (*p* < 0.05), likely due to leaching during soaking and cooking and potential heat lability. Among cooked pulse groups, significantly higher phenolic content was seen in bean varieties (0.90–1.15 mg GAE/g) compared to pea varieties (0.56–0.62 mg GAE/g). This was also reflected in levels of antioxidant activity. These values are lower than published values for cooked cranberry beans (4.26 mg GAE/g DW) and dark red kidney beans (3.11 mg GAE/g DW) analyzed by Padhi et al. [77]; however, in this referenced study, the cooking water was completely evaporated and not discarded as was performed in the present study. Values by Padhi et al. are similar to TPC of unprocessed beans in the current study (2.52–3.15 mg GAE/g DW) and unprocessed regular (darkening) cranberry beans (2.82–4.15 mg GAE/g DW) studied by Chen et al. [29], which further highlights the significance of phenolic loss in cooking water. However, TPC values are in keeping with those reported by Singh et al. for cowpea (1.07 mg GAE/g DW), pigeon pea (0.79–1.21 mg GAE/g DW), and kidney bean varieties, which ranged from 0.25 to 35.11 mg GAE/g DW [20], again demonstrating the variability among strains of similar varieties of pulses.

Trends in antioxidant activity as determined by FRAP, DPPH, and ORAC assays closely followed TPC, consistent with reported trends in the literature [29,77]. Xu et al. [78] reported a strong relation between phenolics and antioxidant activity, indicating that the overall antioxidant activity exerted by food legumes is dominated by phenolic compounds. Again, Padhi et al. demonstrated that TPC was significantly correlated with DPPH (*r* = 0.688, *p* = 0.006), FRAP (*r* = 0.881, *p* < 0.001), and ORAC (*r* = 0.859, *p* < 0.001) [77]. Similarly, Chen et al. reported that antioxidant activity as determined by DPPH, FRAP, and ORAC showed strong positive correlation with TPC (*r*^2^ = 0.99–1.00) [29].

Of note within the results of the present study are the trends in TPC and phenolic activity among *P. vulgaris* L. pulse species. Despite Miss Kelly being considered a variety of the red kidney bean, the phenolic data generated in this study suggest a closer relationship with the cranberry bean being investigated. Visually there is a close resemblance between the seed coats of Miss Kelly and cranberry ns with streaks and patches of red against a lighter red or cream background. The literature points to significantly higher concentrations of flavonoids and anthocyanins in the seed coats of cranberry beans [29,79,80], which may also be true of Miss Kelly and account for these results observed in the present study. Pulses rank high on the list of commonly consumed foods rich in antioxidant activity, and support of their increased consumption in Caribbean diets is recommended given the link of phenolics to reduced risk of many chronic diseases, including cancer, cardiovascular diseases, and diabetes [81].

## 4. Conclusions

This work investigated the effect of cooking on in vitro digestibility, protein quality, as well as starch profile and phenolic content of pulses commonly consumed in the Caribbean diet, acquired from Canada and Jamaica. The protein content of pulses studied ranged from 20 to 26%. As expected, cooking increased protein digestibility as evidenced by various IVPD assays. However, significant differences were noted in the digestibility coefficients when IVPD assays were compared. Of the multi-enzyme assays (PHD and PHS), IVPD as determined by the PHD Equation (4) (PHD4) appears to be most appropriate for pulses, with the strongest correlation to previously published in vivo-derived digestibility coefficients. However, as the regression equations used to determine IVPD from PHD and PHS assays are reliant on a static value in their equations, they cannot accurately identify proteins with low digestibility, as their IVPD equations are restricted to a minimum digestibility ranging from 65.7 to 78.6%. Regardless, while regression equations have been specified for plant-based proteins, the development of pulse protein-specific regression equations merits consideration. While the TSD method holds promise as a static method that more closely simulates the physiological stages of digestion and is also capable of determining a wide range of possible digestibility coefficients, the present protocol requires further optimization to reduce inter-run variability. Cooked cowpeas had the highest amino acid score and the highest average IVPDCAAS across assays. Cooking decreased the percentage of RS while increasing RDS and SDS fractions. Notably, RS and SDS fractions accounted for approximately 20–25% of the total starch content of cooked pulses, which has beneficial implications for cardiometabolic and gut health. The total phenolic content and antioxidant activities were within the expected range for pulse flours, although detailed characterization of phenolics in these pulses is warranted. The results of this work demonstrate that commonly consumed pulses in the Caribbean diet are good sources of protein while also potentially conferring health benefits based on their starch profile, phenolic content, and activity. These findings support recommendations for their increased consumption in the Caribbean diet. Further research is necessary for the improvement and development of in vitro protein quality assays, in particular validated assays to accurately measure in vitro DIAAS in keeping with expert recommendations to transition to this measure from PDCAAS.

## Figures and Tables

**Table 1 foods-14-00283-t001:** Proximate analysis of six varieties of unprocessed and cooked pulse flours as a percentage of dry weight.

	% Dry Matter	% Protein	% Fat	% Total Starch
** Unprocessed **				
** Beans **				
Cranberry Bean	88.32 ± 0.07 ^aA^	23.05 ± 0.21 ^aA^	1.81 ± 0.05 ^aA^	38.54 ± 0.59 ^aA^
Miss Kelly Kidney Bean	87.31 ± 0.04 ^aB^	23.49 ± 0.20 ^aA^	1.79 ± 0.07 ^aA^	35.57 ± 1.21 ^aB^
Red Kidney Bean	89.10 ± 0.04 ^aC^	25.72 ± 0.28 ^aB^	1.42 ± 0.04 ^aB^	33.33 ± 0.83 ^aC^
** Peas **				
Cowpea	89.89 ± 0.01 ^aD^	26.64 ± 0.07 ^aB^	1.42 ± 0.05 ^aA^	45.00 ± 0.60 ^aD^
Pigeon Pea	88.91 ± 0.06 ^aC^	20.13 ± 0.14 ^aC^	1.81 ± 0.06 ^aC^	42.75 ± 0.23 ^aE^
** Cooked **				
** Beans **				
Cranberry Bean	98.13 ± 0.06 ^bA^	22.90 ± 0.48 ^aA^	2.25 ± 0.06 ^bA^	45.14 ± 0.60 ^bA^
Miss Kelly Kidney Bean	97.70 ± 0.08 ^bB^	23.84 ± 0.50 ^aA^	1.84 ± 0.07 ^aB^	41.62 ± 0.65 ^bB^
Red Kidney Bean	97.63 ± 0.23 ^bB^	25.59 ± 0.40 ^aB^	1.73 ± 0.01 ^bB^	42.61 ± 0.81 ^bB^
** Peas **				
Cowpea	97.17 ± 0.04 ^bC^	26.94 ± 0.41 ^aC^	2.03 ± 0.00 ^bC^	51.11 ± 0.45 ^bC^
Pigeon Pea	97.13 ± 0.09 ^bC^	20.54 ± 0.52 ^aD^	2.41 ± 0.06 ^bD^	46.71 ± 0.61 ^bD^

Values are represented as mean ± SD, *n* = 3. Proximate composition was analyzed via one-way ANOVA with Tukey’s HSD test. Means followed by different letters (small in the same pulse class and large in the same treatment) indicate a significant difference between samples (*p* < 0.05).

**Table 2 foods-14-00283-t002:** In vitro protein digestibility (IVPD) of pulse flours as determined by three different assays and calculated using different regression equations.

	pH-Drop–IVPD				pH-Stat–IVPD		Two-Step Digest ^ǂ^
	PHD1	PHD2	PHD3	PHD4 *	PHS1	PHS2	IVPD
**Unprocessed**							
**Beans**							
Cranberry Bean	84.57 ± 0.95 ^aA^	83.83 ± 0.93 ^aA^	85.53 ± 0.83 ^aA^	73.73 ± 0.74 ^aA^	78.03 ± 0.95 ^aA^	81.03 ± 0.78 ^aA^	54.71 ± 3.16 ^aA^
Miss Kelly Kidney Bean	83.87 ± 0.47 ^aA^	83.16 ± 0.47 ^aA^	84.93 ± 0.42 ^aA^	73.20 ± 0.36 ^aA^	77.87 ± 0.29 ^aA^	80.93 ± 0.23 ^aA^	47.27 ± 3.07 ^aA^
Red Kidney Bean	84.00 ± 0.52 ^aA^	83.30 ± 0.52 ^aA^	85.07 ± 0.46 ^aA^	73.33 ± 0.40 ^aA^	77.87 ± 0.29 ^aA^	80.93 ± 0.23 ^aA^	42.28 ± 10.25 ^aA^
**Peas**							
Cowpea	87.40 ± 1.57 ^aA^	86.70 ± 1.57 ^aA^	87.93 ± 1.37 ^aA^	76.03 ± 1.27 ^aA^	80.67 ± 1.33 ^aA^	83.33 ± 1.10 ^aA^	60.42 ± 1.68 ^aA^
Pigeon Pea	84.93 ± 0.67 ^aA^	84.23 ± 0.67 ^aA^	85.87 ± 0.61 ^aA^	74.03 ± 0.57 ^aA^	79.07 ± 0.85 ^aA^	81.93 ± 0.75 ^aA^	52.41 ± 9.62 ^aA^
**Cooked**							
**Beans**							
Cranberry Bean	98.87 ± 1.46 ^bAB^	98.17 ± 1.46 ^bAB^	97.70 ± 1.23 ^bAB^	85.27 ± 1.19 ^bAB^	97.43 ± 1.50 ^bA^	97.77 ± 1.27 ^bA^	72.94 ± 3.54 ^aA^
Miss Kelly Kidney Bean	96.03 ± 1.60 ^bA^	95.33 ± 1.60 ^bA^	95.30 ± 1.35 ^bA^	83.00 ± 1.25 ^bA^	91.47 ± 2.15 ^bB^	92.57 ± 1.85 ^bB^	69.40 ± 7.74 ^aA^
Red Kidney Bean	99.40 ± 0.56 ^bAB^	98.70 ± 0.56 ^bAB^	98.17 ± 0.45 ^bAB^	85.70 ± 0.46 ^bAB^	97.17 ± 1.27 ^bA^	97.50 ± 1.04 ^bA^	60.52 ± 19.36 ^aA^
**Peas**							
Cowpea	99.80 ± 1.04 ^bB^	99.10 ± 1.04 ^bB^	98.50 ± 0.87 ^bB^	86.00 ± 0.87 ^bB^	95.23 ± 1.81 ^bA^	95.90 ± 1.57 ^bA^	68.10 ± 7.98 ^aA^
Pigeon Pea	97.77 ± 1.11 ^bAB^	97.07 ± 1.11 ^bAB^	96.77 ± 1.01 ^bAB^	84.37 ± 0.91 ^bAB^	98.43 ± 0.46 ^bA^	98.63 ± 0.46 ^bA^	70.93 ± 4.62 ^aA^

Values are represented as mean ± SD; IVPD values were analyzed via one-way ANOVA with Tukey’s HSD test. Means followed by different letters (small in the same pulse class and large in the same treatment) indicate a significant difference between samples (*p* < 0.05). pH-Drop, *n* = 3 for unprocessed and *n* = 5 for cooked pulse samples; pH-Stat, *n* = 3 for unprocessed and *n* = 4 for cooked pulse samples; two-step digest, TSD, *n* = 3 for unprocessed and cooked samples. Significant differences (*p* < 0.05) within * and between ^ǂ^ assays were determined by the Robust test for equality of means with post hoc analysis by the Games–Howell test. IVPD formulae: PHD1 74.51 + (22.51 × ΔpH), PHD2 − 73.75 + (22.56 × ΔpH), PHD3 − 76.97 + (19.16 × ΔpH), PHD4 − 65.66 + (18.10 × ΔpH); PHS1 − 76.14 + (47.77 × ΔVol), PHS2 − 78.61 + (40.29 × ΔVol).

**Table 3 foods-14-00283-t003:** Amino acid profiles of cooked and uncooked pulse flours.

	ASP	THR	SER	GLU	PRO	GLY	ALA	CYS	VAL	MET	ILE	LEU	TYR	PHE	LYS	HIS	ARG	TRP
**Unprocessed**																		
**Beans**																		
Cranberry Bean	2.47	0.91	1.08	3.32	0.74	0.84	0.90	0.25	1.12	0.28	0.98	1.67	0.62	1.18	1.51	0.61	1.19	0.19
Miss Kelly Kidney Bean	2.58	0.95	1.04	3.17	0.75	0.88	0.91	0.26	1.19	0.26	1.03	1.68	0.67	1.22	1.51	0.60	1.26	0.18
Red Kidney Bean	2.85	0.96	1.19	3.79	0.81	0.93	0.97	0.24	1.28	0.30	1.11	1.86	0.67	1.36	1.68	0.70	1.34	0.18
**Peas**																		
Cowpea	2.76	0.89	1.05	4.07	0.95	0.96	1.00	0.25	1.22	0.35	1.05	1.79	0.69	1.32	1.66	0.77	1.73	0.24
Pigeon Pea	1.73	0.66	0.78	3.72	0.81	0.67	0.80	0.31	0.83	0.25	0.70	1.27	0.50	1.92	1.27	0.66	1.03	0.12
**Cooked**																		
**Beans**																		
Cranberry Bean	2.76	1.02	1.23	3.51	0.89	0.95	1.03	0.24	1.39	0.31	1.20	1.98	0.73	1.43	1.74	0.69	1.30	0.30
Miss Kelly Kidney Bean	2.89	1.15	1.31	3.37	0.95	1.03	1.08	0.26	1.42	0.30	1.22	2.04	0.77	1.48	1.79	0.69	1.36	0.39
Red Kidney Bean	3.14	1.09	1.35	4.05	1.02	1.06	1.12	0.23	1.54	0.32	1.35	2.23	0.82	1.63	1.95	0.79	1.48	0.33
**Peas**																		
Cowpea	3.11	1.02	1.17	4.64	1.18	1.09	1.18	0.25	1.51	0.44	1.31	2.20	0.82	1.63	1.99	0.89	1.97	0.34
Pigeon Pea	2.06	0.78	0.93	3.76	0.99	0.80	0.97	0.33	1.06	0.30	0.91	1.63	0.59	1.76	1.54	0.78	1.30	0.16

Amino acid composition data are presented as g/100 g flour on an as-is basis.

**Table 4 foods-14-00283-t004:** Amino acid scores of unprocessed and cooked pulse flours.

Amino Acid Score									
	His	Ile	Leu	Lys	Met + Cys	Phe + Tyr	Thr	Trp	Val
**Unprocessed**									
**Beans**									
Cranberry Bean	1.39	1.52	1.10	1.13	0.92	1.24	1.16	**0.75**	1.39
Miss Kelly Kidney Bean	1.34	1.57	1.08	1.11	0.89	1.28	1.19	**0.70**	1.45
Red Kidney Bean	1.43	1.54	1.10	1.13	0.84	1.25	1.10	**0.64**	1.42
**Peas**									
Cowpea	1.52	1.41	1.02	1.07	0.90	1.20	0.98	**0.82**	1.31
Pigeon Pea	1.73	1.24	0.96	1.09	1.11	1.91	0.96	**0.54**	1.18
**Cooked**									
**Beans**									
Cranberry Bean	1.59	1.87	1.31	1.31	**0.96**	1.50	1.31	1.19	1.73
Miss Kelly Kidney Bean	1.52	1.83	1.30	1.29	**0.94**	1.50	1.42	1.49	1.70
Red Kidney Bean	1.63	1.88	1.32	1.31	**0.86**	1.52	1.25	1.17	1.72
**Peas**									
Cowpea	1.74	1.74	1.24	1.27	**1.02**	1.44	1.11	1.15	1.60
Pigeon Pea	2.00	1.58	1.20	1.29	1.23	1.82	1.12	**0.71**	1.47

Bolded values indicate the first limiting amino acid. The FAO/WHO recommended reference pattern used to calculate the amino acid scores was as follows (mg/g protein): Thr—34, Val—35, Met + Cys—25, Ile—28, Leu—66, Phe + Tyr—63, His—19, Lys—58, and Trp—11 [8].

**Table 5 foods-14-00283-t005:** In vitro protein digestibility corrected amino acid score (IVPDCAAS) of pulse flours calculated by various protein digestibility assays.

		PHD IVPDCAAS				PHS IVPDCAAS		TSD IVPDCAAS
	AAS	PHD1	PHD2	PHD3	PHD4	PHS1	PHS2	
**Unprocessed**								
**Beans**								
Cranberry Bean	0.75	0.63	0.63	0.64	0.55	0.59	0.61	0.41
Miss Kelly Kidney Bean	0.70	0.59	0.58	0.59	0.51	0.55	0.57	0.33
Red Kidney Bean	0.64	0.54	0.53	0.54	0.47	0.50	0.52	0.27
**Peas**								
Cowpea	0.82	0.72	0.71	0.72	0.62	0.66	0.68	0.50
Pigeon Pea	0.54	0.46	0.45	0.46	0.40	0.43	0.44	0.28
**Cooked**								
**Beans**								
Cranberry Bean	0.96	0.95	0.94	0.93	0.82	0.93	0.94	0.70
Miss Kelly Kidney Bean	0.94	0.90	0.90	0.90	0.78	0.86	0.87	0.65
Red Kidney Bean	0.86	0.85	0.84	0.84	0.73	0.83	0.84	0.52
**Peas**								
Cowpea	1.02	1.02	1.01	1.00	0.88	0.97	0.98	0.69
Pigeon Pea	0.71	0.69	0.68	0.68	0.59	0.69	0.70	0.50

IVPDCAAS was calculated as the product of AAS and IVPD as determined by the following regression equations: PHD1 − 74.51 + (22.51 × ΔpH), PHD2 − 73.75 + (22.56 × ΔpH), PHD3 − 76.97 + (19.16 × ΔpH), PHD4 − 65.66 + (18.10 × ΔpH); PHS1 − 76.14 + (47.77 × ΔVol), and PHS2 − 78.61 + (40.29 × ΔVol).

**Table 6 foods-14-00283-t006:** Starch digestibility profile of five varieties of unprocessed and cooked pulse flours.

Starch Digestibility					
	By Megazyme K-DSTRS			By Megazyme K-RSTAR	
	RDS	SDS	RS_4h_	TDS	RS_16h_
**Unprocessed**					
**Beans**					
Cranberry Bean	3.51 ± 0.24 ^aA^	7.05 ± 0.97 ^aA^	89.44 ± 1.03 ^aA^	12.45 ± 3.66 ^aA^	87.55 ± 3.66 ^aA^
Miss Kelly Kidney bean	3.61 ± 0.22 ^aA^	7.29 ± 1.27 ^aA^	89.10 ± 1.44 ^aA^	10.85 ± 1.23 ^aAB^	89.15 ± 1.23 ^aAB^
Red Kidney Bean	4.84 ± 0.43 ^aA^	8.70 ± 2.86 ^aA^	86.45 ± 3.07 ^aAB^	8.17 ± 3.64 ^aB^	91.83 ± 3.64 ^aB^
**Peas**					
Cowpea ^¥^	12.86 ± 0.52 ^aB^	40.47 ± 2.12 ^aB^	46.67 ± 2.42 ^aC^	97.24 ± 0.23 ^aC^	2.76 ± 0.23 ^aC^
Pigeon Pea ^¥^	3.32 ± 0.38 ^aA^	12.98 ± 3.72 ^aC^	83.70 ± 3.37 ^aB^	34.10 ± 2.51 ^aD^	65.90 ± 2.51 ^aD^
**Cooked**					
**Beans**					
Cranberry Bean ^¥^	75.89 ± 2.09 ^bA^	11.03 ± 0.59 ^bAC^	13.08 ± 2.48 ^bA^	90.83 ± 0.33 ^bA^	9.17 ± 0.33 ^bA^
Miss Kelly Kidney Bean	71.72 ± 0.99 ^bB^	20.87 ± 2.34 ^bB^	7.40 ± 2.51 ^bBC^	90.21 ± 0.19 ^bA^	9.79 ± 0.19 ^bA^
Red Kidney Bean	79.47 ± 2.35 ^bCD^	9.83 ± 1.18 ^aACD^	10.70 ± 1.87 ^bAB^	90.23 ± 0.08 ^bA^	9.77 ± 0.08 ^bA^
**Peas**					
Cowpea ^¥^	76.71 ± 2.74 ^bAC^	12.97 ± 0.98 ^bC^	10.32 ± 2.59 ^bAB^	96.02 ± 0.04 ^aB^	3.98 ± 0.04 ^aB^
Pigeon Pea ^¥^	81.41 ± 1.41 ^bD^	13.42 ± 1.56 ^aC^	5.17 ± 2.78 ^bC^	91.27 ± 0.23 ^bA^	8.73 ± 0.23 ^bA^

Data are expressed on a dry weight basis as a percentage of total starch. Values are expressed as mean ± SD, where *n* = 3; RDS, rapidly digestible starch; SDS, slowly digestible starch; TDS, total digestible starch, starch digested within 16 h; RS, resistant starch; RS_4h_, starch not hydrolyzed after 4 h; RS_16h_, starch not hydrolyzed after 16 h. Starch digestibility fractions were analyzed via one-way ANOVA with Tukey’s HSD test. Means followed by different letters (small in the same pulse class and large in the same treatment) indicate a significant difference between samples (*p* < 0.05). ¥ A significant difference (*p* < 0.05) between resistant starch values obtained by K-DSTRS (RS_4h_) vs. K-RSTAR (RS_16h_) was determined by paired *t*-test.

**Table 7 foods-14-00283-t007:** Total phenolic contents and phenolic activities of six varieties of unprocessed and cooked pulse flours.

	TPC	FRAP	DPPH	ORAC
	(mg GAE/g)	(µmol AAE/g)	(µmol TE/g)	(µmol TE/g)
** Unprocessed **				
** Beans **				
Cranberry Bean	3.09 ± 0.11 ^aA^	33.46 ± 0.90 ^aA^	29.07 ± 0.94 ^aA^	461.51 ± 7.52 ^aA^
Miss Kelly Kidney Bean	3.15 ± 0.10 ^aA^	35.44 ± 1.60 ^aB^	28.44 ± 0.94 ^aA^	463.91 ± 28.27 ^aA^
Red Kidney Bean	2.52 ± 0.08 ^aB^	30.48 ± 0.51 ^aC^	24.19 ± 1.18 ^aB^	415.86 ± 15.30 ^aB^
** Peas **				
Cowpea	1.31 ± 0.07 ^aC^	19.20 ± 0.17 ^aD^	9.89 ± 0.48 ^aC^	223.61 ± 9.23 ^aC^
Pigeon Pea	0.94 ± 0.02 ^aD^	13.82 ± 0.23 ^aE^	6.43 ± 0.05 ^aD^	153.78 ± 7.64 ^aD^
** Cooked **				
** Beans **				
Cranberry Bean	1.15 ± 0.02 ^bA^	12.93 ± 0.32 ^bA^	8.55 ± 0.29 ^bA^	77.24 ± 2.76 ^bA^
Miss Kelly Kidney Bean	1.14 ± 0.01 ^bA^	13.38 ± 0.17 ^bA^	8.72 ± 0.43 ^bA^	71.72 ± 2.19 ^bAB^
Red Kidney Bean	0.90 ± 0.01 ^bB^	11.22 ± 0.16 ^bB^	7.04 ± 0.09 ^bA^	63.08 ± 1.51 ^bAB^
** Peas **				
Cowpea	0.62 ± 0.01 ^bC^	6.75 ± 0.11 ^bC^	3.59 ± 0.12 ^bB^	52.15 ± 2.05 ^bBC^
Pigeon Pea	0.56 ± 0.02 ^bC^	7.07 ± 0.07 ^bC^	3.78 ± 0.11 ^bB^	39.37 ± 1.44 ^bC^

Values are represented as mean ± SD, where *n* = 3. Phenolic content and phenolic activity were analyzed via one-way ANOVA with Tukey’s HSD test. Means followed by different letters (small in the same pulse class and large in the same treatment) indicate a significant difference between samples (*p* < 0.05). TPC, total phenolic content; FRAP, ferric reducing ability of plasma; DPPH, 2,2-diphenyl-1-picrylhydrazyl; ORAC, oxygen radical absorbance capacity.

## Data Availability

The original contributions presented in this study are included in the article; further inquiries can be directed to the corresponding authors.

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
