# Peer review of "Assessment of Protein Quality and Nutritional Characteristics of Commonly Consumed Pulses in the Caribbean Diet by Different In Vitro Assays"

_foods, 2025, doi:10.3390/foods14020283_

Round 1

Reviewer 1 Report

Comments and Suggestions for Authors

This manuscript focused on investigating the protein nutritional properties considering the Caribbean diet. However, the manuscript should be revised before. Some further comments are as follows:

For the key words, ‘IVPD; PDCAAS; PER’, full spelling is necessary.

Please be careful for the grammar errors, like line 24 ‘digest assay..’.

How many times for each the detection?

For the data in table, please show the right form for the standard deviation using ±.

Comments on the Quality of English Language

This manuscript focused on investigating the protein nutritional properties considering the Caribbean diet. However, the manuscript should be revised before possible publication. Some further comments are as follows:

For the key words, ‘IVPD; PDCAAS; PER’, full spelling is necessary.

Please be careful for the grammar errors, like line 24 ‘digest assay..’.

How many times for each the detection?

For the data in table, please show the right form for the standard deviation using ±.

Author Response

Comments on the Quality of English Language

This manuscript focused on investigating the protein nutritional properties considering the Caribbean diet. However, the manuscript should be revised before possible publication. Some further comments are as follows:

 Comment: For the key words, ‘IVPD; PDCAAS; PER’, full spelling is necessary.

Thank you for this comment. We have rewritten IVPD and in vitro protein digestibility. However, as PDCAAS and PER are commonly used terms in the sphere of protein quality research, we have continued to include those in the keywords as they provide more reliable indexing and reader accessibility.

Please be careful for the grammar errors, like line 24 ‘digest assay..’.

Thank you for highlighting this. “two-step digest assay” has been rewritten as “two-step digestibility assay”.

How many times for each the detection?

Thank you for this question. Regarding pulse flours, samples were analysed in triplicate (n=3). The exception to this was pH Drop and pH Stat, with an n=5 and n=4 for cooked pulse samples respectively. These have been indicated in the respective table footnotes.

For the data in table, please show the right form for the standard deviation using ±.

Thank you for this recommendation. Data tables have been updated in the format of “mean ± SD” as suggested.

Reviewer 2 Report

Comments and Suggestions for Authors

The manuscript describes the effect of home cooking on pulses, often used in the Carribean kitchen. It covers several analytic methods and several varieties of pulses. Still, the overall purpose could be more clear, eg do you aim for comparing the different methods of IVPD or is the main interest comparison of the different types of pulses? Furthermore, only include one way of cook the pulses also limits the interest of the results, as we do already know that changes occur during boiling. It is also unclear to me, of the ‘end use’ of the legumes are the flour or the beans themselves. I don’t think you would ever eat the pulses uncooked and therefore the comparison of boiled- non-boiled is of less importance. Please make the aim more clear and also extend the discussion accordingly.

Some specific comments:

line 117 – how long time were they cooked? It is not sufficient to write ‘ until they were done’

line 243 – why did you chose a one-way ANOVA and not a two way? (Cooking and variety being the two factors)?

How many replicates did you perform?

It would have been interesting, of you had run a BSA IVPD as well as a control of the methodology

Why did you chose the FAO/WHO requirement for children?

Table 3 is difficult to read. Can it be made horizontal? And please make the table text more descriptive eg include the units.

Table 4. Please make the limiting amino acid in bold. In general they are very high compared to what is usually seen in pulses. Please comment on that.

As the pulses were purchased in a shop, and as far as I can read, only one purchase was done, you miss some information about the variation within each variety. This is a weakness when you want to compare the different varieties and should be discussed as such in the end of the manuscript.

Author Response

The manuscript describes the effect of home cooking on pulses, often used in the Carribean kitchen. It covers several analytic methods and several varieties of pulses. Still, the overall purpose could be more clear, eg do you aim for comparing the different methods of IVPD or is the main interest comparison of the different types of pulses?

Furthermore, only include one way of cook the pulses also limits the interest of the results, as we do already know that changes occur during boiling.

It is also unclear to me, of the ‘end use’ of the legumes are the flour or the beans themselves. I don’t think you would ever eat the pulses uncooked and therefore the comparison of boiled- non-boiled is of less importance. Please make the aim more clear and also extend the discussion accordingly.

Thank you for these comments.  We have revised the introduction to better highlight the purpose of this work.  As mentioned, it has been established that boiling will alter the nutritional characteristics of pulses such as beans and lentils, however less clear is the impact of different in vitro methods of assessment on nutritional characterization.

Some specific comments:

line 117 – how long time were they cooked? It is not sufficient to write ‘until they were done’

Thank you for your comment. Approximate cook time for the seeds was between 25-35 minutes, and has now been included. As we prepared several varieties of pulse seeds, we found that they had varied cooking times. As such for further descriptive purposes we qualified the end of this cook-time as “being easily compressible between thumb and index finger”. This descriptor has been used in other publications in which multiple varieties of pulse seeds have been prepared, such as:

Nosworthy, M.G.; Medina, G.; Franczyk, A.; Neufeld, J.; Appah, P.; Utioh, A.; Frohlich, P.; House, J.D. Effect of processing on the in vitro and in vivo protein quality of red and green lentils (Lens culinaris). Food Chem. 2017, 240, 588–593.

Baeza-Jiménez, Ramiro, and Leticia X. López-Martínez. "Changes in Phenolic Composition and Bioactivities of Ayocote Beans under Boiling (Phaseolus coccineus L.)." Molecules 29, no. 16 (2024): 3744.

line 243 – why did you chose a one-way ANOVA and not a two way? (Cooking and variety being the two factors)?

Thank you for this question. One-Way ANOVA was selected to identify significant differences rather than a Two-Way ANOVA as there would be little value in comparing the results from unprocessed beans to processed peas or unprocessed peas to processed beans.

How many replicates did you perform?

Response: Thank you for this question. As it pertains to replicates, samples were analysed in triplicate (n=3). The exception to this was pH Drop and pH Stat, with an n=5 and n=4 for cooked pulse samples respectively. These have been indicated in the respective table footnotes.

It would have been interesting, of you had run a BSA IVPD as well as a control of the methodology

“A casein control was included in each experimental run as a control sample.” This sentence has now been included in sections 2.6 and 2.7.

Why did you chose the FAO/WHO requirement for children?

Thank you for this question. In U.S. regulations, the pre-school children (2–5 years) amino acid scoring pattern (FAO/WHO, 1991) is the currently required reference (FDA, 2018) for establishing protein content claims and determining PDCAAS values. Although it is the reference pattern for 2-5 years it has been implemented in the United States as the required method for protein quality determination and evaluation of the eligibility of foods for protein content claim purposes targeted at children through adults (FAO/WHO, 1991).

U.S. Food and Drug Administration and Department of Health and Human Services, 2018. Nutrition labeling of food. 21 CFR, Part 101, Subpart A, Section 101.9. Electronic Code of Federal Regulations; Government Printing Office. (https://www.ecfr.gov/current/title-21/chapter-I/subchapter-B/part-101/subpart-A/section101.9)

Table 3 is difficult to read. Can it be made horizontal? And please make the table text more descriptive eg include the units.

Thank you for highlighting this. We agree that the table is difficult to read and will confer with the publisher regarding an appropriate solution. Units have been placed in the footnotes.

Table 4. Please make the limiting amino acid in bold. In general they are very high compared to what is usually seen in pulses. Please comment on that.

Thank you for this suggestion. Yes, this has been placed in bold. Regarding the high AAS for the limiting amino acids, yes we agree that they are high, however they are comparable to data for AAS in the literature. This has been expounded upon in the results and discussion Section 3.3.

As the pulses were purchased in a shop, and as far as I can read, only one purchase was done, you miss some information about the variation within each variety. This is a weakness when you want to compare the different varieties and should be discussed as such in the end of the manuscript.

Thank you for this comment. Due to the nature of this experiment, a direct comparison of varietal variation was not a planned outcome.  This important comparison will be incorporated into upcoming experiments that build from this initial work.

Reviewer 3 Report

Comments and Suggestions for Authors

Manuscript: Effect of a domestic cooking method on in vitro protein quality and
other nutritional characteristics of commonly eaten pulses in the Caribbean diet. Thomas et al

According to the title and the text, the aim of this manuscript should be to investigate the effect of the cooking method on some pulses. However, the authors have considered only one type of seed processing and analysed several parameters with two or three methods discussing the collected results.  This is not in agreement with the title neither or with the declared aim.

Moreover, the authors reported more time in the manuscript that have analysed the pulse flour.  Also this is not correct either they analysed raw and cooked seeds which, of course, must be milled before the biochemical analyses.

Lines 46-48. The sentence does not reflect the analysed materials, as they were not local Caribbean varieties (only kidney bean is declared to be a local varieties) and some samples were acquired in a Canadian supermarket (lines 111-112). What significance could be attributed to data relative to materials not cultivated in the same environment and in the same year?

The description of methods should be reduced being they available in the literature. If significant modifications to the described methods were made, authors should explain the reasons for the changes and demonstrate the reproducibility of the modified procedures.

The discussion should be improved. For example, I do not see differences in fat for Kelly bean cooked and unprocessed.

Minor remarks

Lines 39-44 add author to scientific name of species

Line 53 only protease inhibitors have a significant effect on protein digestibility

Can the use of distilled water be considered a home cooking method?

Author Response

Manuscript: Effect of a domestic cooking method on in vitro protein quality and
other nutritional characteristics of commonly eaten pulses in the Caribbean diet. Thomas et al

According to the title and the text, the aim of this manuscript should be to investigate the effect of the cooking method on some pulses. However, the authors have considered only one type of seed processing and analysed several parameters with two or three methods discussing the collected results.  This is not in agreement with the title neither or with the declared aim.

Thank you for this comment. The title  and aims have been revised to be more congruent with the aim to read as follows, “Assessment of protein quality and nutritional characteristics of commonly consumed pulses in the Caribbean diet by different in vitro assays”.

Moreover, the authors reported more time in the manuscript that have analysed the pulse flour.  Also this is not correct either they analysed raw and cooked seeds which, of course, must be milled before the biochemical analyses.

Thank you for your comment. Pulses were milled into flours prior to all analyses.

Lines 46-48. The sentence does not reflect the analysed materials, as they were not local Caribbean varieties (only kidney bean is declared to be a local varieties) and some samples were acquired in a Canadian supermarket (lines 111-112). What significance could be attributed to data relative to materials not cultivated in the same environment and in the same year?

Thank you for this comment. The nature of this experiment was not looking at genotype/ environmental/ year differences among pulse cultivars. Additionally, it should be noted that the concept was to analyse pulses commonly consumed in the Caribbean diet, not to be confused with pulses cultivated in the Caribbean. This distinction is important as when one looks at the data regarding importation of crops, the greater quantity of pulses consumed in the Caribbean are obtained from international suppliers, including Canada. It should also be noted that the Caribbean diaspora continue to consume Caribbean-influenced meals prepared in similar manner as those residing in the Caribbean.

The description of methods should be reduced being they available in the literature. If significant modifications to the described methods were made, authors should explain the reasons for the changes and demonstrate the reproducibility of the modified procedures.

Thank you for your recommendation. The methods have been reduced, in particular sections 2.7 In vitro protein digestibility by two-step static enzymatic digestibility assay and Determination of total polyphenol content (TPC) and antioxidant activity of pulse flours has been combined into section 2.9.

The discussion should be improved. For example, I do not see differences in fat for Kelly bean cooked and unprocessed.

Thank you for highlighting this discrepancy. We have reviewed the data tables and corrected these statistics. The discussion has been updated based on comments provided by yourself as well as the other reviewers.

Minor remarks

Lines 39-44 add author to scientific name of species

Thank you for your comment.  We have modified the body of the text to include ‘L.’ where appropriate.

Line 53 only protease inhibitors have a significant effect on protein digestibility

Thank you for this comment. While protease inhibitors do indeed have a direct effect on protein digestibility, there is a wide body of evidence to support the effect of these other anti-nutritional factors such as tannins and phytic acid hemagglutinins on protein digestibility

G Sarwar Gilani, Kevin A Cockell, Estatira Sepehr, Effects of Antinutritional Factors on Protein Digestibility and Amino Acid Availability in Foods, Journal of AOAC INTERNATIONAL, Volume 88, Issue 3, 1 May 2005, Pages 967–987, https://doi.org/10.1093/jaoac/88.3.967

Sarwar Gilani G, Wu Xiao C, Cockell KA. Impact of Antinutritional Factors in Food Proteins on the Digestibility of Protein and the Bioavailability of Amino Acids and on Protein Quality. British Journal of Nutrition. 2012;108(S2):S315-S332. doi:10.1017/S0007114512002371

Yasmin, A., Zeb, A., Khalil, A. W., Paracha, G. M. U. D., & Khattak, A. B. (2008). Effect of processing on anti-nutritional factors of red kidney bean (Phaseolus vulgaris) grains. Food and Bioprocess Technology1(4), 415-419.

Can the use of distilled water be considered a home cooking method?

Thank you for this question. While the use of distilled water would not be considered a home cooking method, we opted for distilled water to reduce any variability in water quality from day-to-day.

Round 2

Reviewer 2 Report

Comments and Suggestions for Authors

The manuscript has been revised accordingly to the comments and can be published in its current form

Reviewer 3 Report

Comments and Suggestions for Authors

Several concerns have been adressed in this new version of the manuscript.